# Why Do Patients with Mental Disorders Default Treatment? A Qualitative Enquiry in Rural Kwazulu-Natal, South Africa

**DOI:** 10.3390/healthcare9040461

**Published:** 2021-04-14

**Authors:** Kebogile Elizabeth Mokwena, Jabulile Ndlovu

**Affiliations:** 1Department of Public Health, Sefako Makgatho Health Sciences University, Pretoria 0204, South Africa; jabulilendlovu5@gmail.com; 2Manguzi Hospital, Kwa Zulu Natal, Pietermaritzburg 3201, South Africa

**Keywords:** South Africa, mental disorders, treatment default, treatment outcomes, rural, qualitative study

## Abstract

Although treatment default by psychiatric patients or mental health care users is a global challenge, this behavior is reported to be higher in South Africa. The Manguzi District Hospital in rural Kwa-Zulu Natal Province, South Africa, experiences high rates of treatment default by psychiatric patients. The objective of this study was to determine the reasons for treatment defaulting at Manguzi Hospital, KwaZulu-Natal Province, South Africa. An explorative qualitative design, using in-depth interviews, was conducted with mental health care users who had defaulted out-patient psychiatric treatment. Twenty-one mental health care users were interviewed before data saturation was reached. Nvivo version 11 was used to analyze the qualitative data. Major themes that emerged confirmed that social factors are key contributions to treatment defaulting, and these include denial of the mental disorders; belief that they are cured; lack of, or disintegration of social support; preference for traditional medicine; and flaws in the health care system. Social determinants of treatment outcomes for mental disorders require tailor-made support systems for patients in these rural communities, which include increase in health literacy and attention to the cultural understanding of mental disorders.

## 1. Introduction

South Africa has a high prevalence of chronic mental disorders [1,2], and these can be partially explained by the political history and the current unfavorable social conditions [3], particularly material inequalities [4]. In the face of inadequate resources for treatment of mental disorders and promotion of mental health [5], there is a need to identify barriers to effective delivery of mental health services at all levels. 

Adherence to treatment is key to the successful management of chronic mental disorders and in the absence of this, relapse is frequent, which compromises the treatment outcomes [6]. Regular scheduled attendance at health facilities is necessary to enable treatment assessments because this is associated with positive treatment outcomes. Treatment defaulting often leads to relapse, which may need hospital re-admission to stabilize the patient before being discharged home as an out-patient. Many chronic mental disorders allow for patients to be treated at home, combined with periodic assessment of their condition and reissue of medication. A break in this process, which often results from treatment default, is a challenge in many health systems, and needs attention if the intended treatment outcomes are to be achieved. 

Although literature on treatment default by mental health care users for developed countries is available, this is not so in developing countries [7,8], including South Africa. The defaulter rate among psychiatric patients in South Africa is reported to be as high as 69%, which is higher than the 44% to 50% range reported globally [9], and twice that reported for other health programs in South Africa.

Many people who have mental illness require a significant amount of assistance and support from family members and/or other carers, for simple responsibilities such as daily self-care and even their appointments at health facilities for evaluation and issuing of medication. The South African Mental Health Act of 2002 advocates the de-institutionalization of treatment of patients with mental disorders, so that the mental health care users are treated mostly in their communities and homes. The special needs to care for such patients are often not met [10], which includes supervision for meeting hospital appointments. Both the supervision and resultant attendance of hospital are required to promote optimum treatment outcomes.

Understanding of mental illness by relatives and/or carers is essential to enable them to support the patient and encourage him/her to honor appointments and adhere to medication. Lack of this understanding results in the relatives of the mental health care user or patient being secretive about the illness, which results in the breaking of the chain of treatment and resultant treatment default [11]. Conversely, treatment acceptability by the patient determines the extent to which the patient will adhere to the treatment, hence the need for efforts to improve acceptability of treatment, which includes incorporating patients’ understanding of the disorder and patient–provider interactions [12].

The proportion of Africans who receive treatment for mental health problems is extremely low [13], and treatment default contributes to the lower numbers that receive treatment. In South Africa, caring for a patient with mental illness comes with additional costs for the family [10,14], which may impact on the ability of the patient to access health facilities. This is particularly true for people of lower economic status, who cannot afford medical insurance and among which treatment facilities are fewer and more difficult to access.

Mental illness-related stigma has been reported to contribute to disempowerment, which translates to treatment default and a barrier to recovery [15]. Mental health stigma discourages individuals from obtaining proper mental health treatment, and the interface of mental illness, stigma, and mental health treatment has ethical and potentially moral implications, which cannot be ignored [16].

Treatment default complicates the envisaged treatment outcomes and frustrates the prevention of relapse, and has significant therapeutic and socio-economic implications [8,17]. Because predictors of treatment default are complex, and service barriers and administrative errors are common, there is a need to conduct studies in various settings to enable responsive interventions that can help mental health care service providers to understand the personal, social, and health-related issues encountered by mental health care users that contribute to treatment defaulting. This can enable the process of setting priorities for both the patient and their family members regarding treatment default prevention and mental health promotion after discharge from hospital.

Thus, efforts towards the improvement of adherence to treatment schedule, including medication, enhances the effectiveness of mental health services. Determining reasons for the high default rate among patients with mental disorders is therefore needed. Hence, this study aimed to determine the reasons for treatment defaulting by patients who have mental disorders and are treated at Manguzi Hospital, KwaZulu-Natal Province, South Africa.

## 2. Methods

An explorative qualitative design, using in-depth interviews, was used to collect data from twenty-one (21) mental health care users/patients with mental disorders who had defaulted their monthly follow-up appointments at Manguzi Hospital, KwaZulu-Natal Province in South Africa. The sample of 21 was determined by content data saturation, in which additional interviews were no longer producing new information.

### 2.1. Study Setting

The study was conducted at Manguzi Hospital, which is in a rural area of Kwa-Zulu Natal Province in South Africa. It is a general hospital with a dedicated mental health unit which caters for both in- and out-patients.

### 2.2. Study Participants

The study population consisted of patients who were diagnosed with mental illness and were receiving treatment from Manguzi hospital and had defaulted treatment. From that population, a purposive sample was determined from the list of both male and female patients who were receiving treatment for either schizophrenia or major depressive disorder, and were identified as treatment defaulters by the community health workers. The classification as a treatment defaulter was determined by missing more than 2 consecutive scheduled appointments at the hospital.

### 2.3. Recruitment of Study Participants

The professional nurse in charge of the psychiatric services at the hospital provided names of patients who were receiving treatment for either schizophrenia or major depressive disorder, and had defaulted their hospital follow-up appointments on more than two occasions (this is practice procedure for patients who default treatment). This list was used by the researcher to trace the potential participants in their homes, and were requested to come to the outpatient psychiatric clinic on given appointment dates, from where they were requested to participate in the study.

### 2.4. Data Collection

Data were collected in an office in the psychiatric department of the hospital. On the day of data collection, the identified patient who had defaulted treatment underwent the prescribed clinical examination by a professional psychiatric nurse to establish the mental health status. If the mental state of the client was deemed ideal, he/she was then referred to the researcher for interview. The purpose of the study was explained and a consent form was read to the participant in isiZulu, which is the indigenous language of the area. The participant was given an opportunity to ask questions and seek any further clarifications. If there were no further questions, the participant was then requested to sign the consent form, which was followed by collection of demographic data and the in-depth interview, using a researcher-developed interview guide. The interviews, which lasted between 20 and 30 min, were voice recorded using digital recorders.

### 2.5. Data Analysis

Demographic data was analyzed descriptively and summarized as percentages and proportions. The digitally recorded data was transcribed verbatim and typed into Word. The typed transcripts were uploaded to NVivo 11 for coding and development of themes. A codebook with definitions for each code was developed from the first transcript. The codes were then applied to all transcripts and the codebook was continually modified as necessary and as new codes emerged from combing through additional transcripts.

## 3. Findings

### 3.1. Sociodemographic Characteristics

The sample consisted of 21 participants, 11 males and 10 females, who had defaulted treatment for their scheduled appointments for at least two months before the study was conducted. Their ages ranged from 19 to 45 years, with a mean of 35 years, and 52% (n = 11) were aged 35 years and above. Most were single with a primary level of education and unemployed, with a few receiving disability grants. The sociodemographic characteristics are shown in Table 1.

### 3.2. Findings from Qualitative Data

The major themes emerged from the qualitative data analysis, in addition to the verbatim quotations that supported the emergence of the themes, are shown below.

### 3.3. Belief That They Are Cured

This theme refers to the views of the patients that they are cured from their mental disorder for which they took treatment for some time, that they no longer need the treatment, and hence the default from further visits to the health center. Such views were expressed as follows:

“I told myself that I was better, and then I decided to stop taking my medication.” (44 year old male with schizophrenia)

“I saw myself as being alright, I was no longer sick, I am healthy now.” (31 year old female with schizophrenia)

“I felt that my body was feeling better, I felt as if I was healed.” (26 year old male with schizophrenia).

### 3.4. Lack of Social Support System

This theme refers to the collapse of the social system which provided support to enable and encourage the patient to adhere to treatment appointments, which was followed by treatment default after the support failed or was withdrawn. This support was provided by a range of people such as family members, friends, and neighbors, and was in the form of reminders about hospital dates and /or being accompanied to the hospital. This theme also included emotional support shown by specific individuals for being concerned about the well-being of the participants. The collapse of such a social support system was expressed as follows:

“I used to be accompanied by my neighbor when my illness was still worse.” (45 year old female with schizophrenia).

“I stopped because my parents separated and my mother, who used to accompany me to hospital to collect pills, left me and married another man.” (25 year old female with major depression)

“My mother, who used to accompany me died and there was no one else who could accompany me.” (43 year old male with schizophrenia).

### 3.5. Mobility of Patients

This theme refers to the common practice of the participants not always having a stable place of abode, and often staying with relatives and friends for unplanned and unspecified periods of time, during which time their medications would be used up and they would not go to get more.

“I was away visiting relatives in Durban. I overstayed. I left my hospital card at home.” (27 year old female with major depression).

“I quarreled with my brother and I decided to cross the border and visit a friend in Mozambique, so that I can cool off and give him some space.” (30 year old male with schizophrenia).

### 3.6. Denial of Mental Disorder

This theme refers to the participants’ denial about the existence or seriousness of their mental disorder, which results in failing to acknowledge the importance of attending the scheduled health appointments. Stigma about mental illness may be contributing to this denial, as shown by the following quotes:

“Some people take me as somebody who is mentally ill, and they do not seriously take my opinions regarding any matter under discussion. Even if my opinion or advice would have been useful, it is disregarded because I am taken as a person who is not thinking straight because I take pills for mental illness.“ (31 year old female with schizophrenia).

“Me, I do not care that I have not come to collect my pills, what I care about is that my mother is dead.” (43 year old male with schizophrenia).

### 3.7. Poor or Inadequate Communication

This theme means that communication between participants and the mental health care workers regarding the management of their mental illness was not adequate. This includes lack of or inadequate communication regarding the prognosis of their mental illness, and the views were expressed as follows:

“The doctor did not say for how long I will be on medication. It is just that I was not told here at the hospital that if I stopped taking the pills on my own decision, I shall be at risk of being disturbed in my mind again.” (31 year old female with schizophrenia)

“The nurses did not explain to me that I would be expected to collect my pills at this clinic. I was told that I will collect my pills at the hospital general OPD.” (45 year old female with major depression).

### 3.8. Disintegrated System of the Hospital

This theme refers to a number of instances in which the disintegrated hospital system failed to support the participants and actually contributed to their treatment default. Such instances included when participants would be admitted to hospital with a different condition, and the medication for mental disorder was not included in the holistic treatment of such a patient.

“I was admitted to hospital for another illness. I did not bring my pills for mental illness and even at the hospital, they forgot to give me my medication, although it was written on my hospital carry-card. I thought that if nurses in hospital do not give me these pills while I am admitted, it means that it is not important to drink them always.” (42 year old female with major depression)

### 3.9. Resorting to Traditional Treatment

This theme refers to situations in which the participants have a preference for traditional treatment for any reason, and opt to replace the health facility treatment with traditional medicine.

“I was happy to be on the pills until my illness got complicated and I had to see a traditional healer. I therefore stopped taking pills because I am still using traditional medicines.” (19 year old male with schizophrenia)

### 3.10. Other Reasons

Additional reasons for treatment default include challenges with staff, such as rude remarks by health personnel; unavailability of medicines at health centers, where patients would not be given all their medications due to unavailability; side effects of the medicines (where patients would experience adverse or uncomfortable side effects); and poor communication by health personnel, which included unclear communication about the nature of their disorders.

## 4. Discussion

Patients with chronic mental disorders need a range of care interventions that will assist in the management of the disease, and, without this care, mild disorders can complicate and worsen to result in great disability [18]. Integral to the care needed by patients who have mental disorders is a support system that will enable them to participate in a range of activities to enhance their recovery, which includes encouragement to attend treatment appointments. This study found that the failing or withdrawal of social support leads to treatment defaulting, whether such support comes from friends, spouse, or relatives [19].

Denial of the existence of a mental disorder is reported to be common among patients with schizophrenia [20], and is an obstacle to receiving appropriate care. Denial hampers treatment outcomes because patients who are in denial are more likely to miss health care appointments, and are likely to experience serial hospitalizations [21]. The study confirms that denial contributed to treatment default in this sample, and until denial is addressed, the likelihood of treatment retention remains low [22]. Although denial of mental disorders (and other serious diseases) is common among patients, it is regarded as self-deception [23], and can be explained by poor insight regarding the nature of the disease [24], and/or is often a coping mechanism for a difficult situation by minimizing it [25]. Denial also plays a major role in addiction [26], because the affected person deals with addiction by dismissing its existence. Moreover, individuals in Sub-Saharan Africa often attribute mental illness to alcohol/illicit drug use and spirit possession [27], which influences treatment seeking behavior regarding both honoring of appointments and adherence to prescribed medicines. 

Among patients diagnosed with schizophrenia, denial is a marker of greater aggressive tendencies [28]. Denial is reported to be common among patients who have low health literacy, and is associated with poorer use of health care services and resultant poorer health outcomes [29]. There are views that because mental health disorders are so prevalent, there is a need to increase the community’s mental health literacy to enable the whole community to take action for better mental health [30]. The expressed belief that they are cured may be an extension of denial as explained above, and may also be linked to a lack of insight into the nature of mental disorder or inadequate communication from health workers regarding the need for chronic treatment.

Some patients diagnosed with mental disorders often move places of residence and do not stay in one area for prolonged periods. This residential mobility interferes with health care receipt [31], and, in this study, it was found to be one of the reasons for missing treatment schedules. Difficulties and/or inability to access appropriate health care when patients are away from home contributes to more serious health concerns [32].

Among Africans, particularly those that find it difficult to understand the complexity of psychiatric illness, the preference for alternative mental health services in the form of African traditional medicines often exists [33]. Although academic health services literature may argue that little evidence exists to suggest that such medicines change the course of mental illnesses, there is some evidence that suggests that traditional healers provide effective psychosocial interventions, which can relieve distress and improve mild mental health symptoms. This is beneficial to, and appreciated by, the patient [34,35]. African traditional medicines also meet cultural expectations of patients that biomedicine does not, and therefore has a positive role to play in the treatment of mental disorders in Africa [36]. Traditional healers therefore seem to play an important role in the delivery of mental health care in South Africa [37]. However, shifting between treatment modalities, as found in this study, contributes to treatment defaulting and compromises both treatment adherence and treatment outcomes [38].

The flaws in the health care system, which resulted in patients defaulting from their mental disorder treatment while admitted to hospital for other conditions, can be solved by observing the standard protocol of interviewing patients, which includes asking patients what they have been treated for. The findings of this study are a reminder of the role of the health system in enhancing treatment compliance of patients, which includes accurate communication to the patients. Treatment default should be prevented by all, and a key strategy to encourage patients to adhere to their health care appointments is to enhance their experience when they visit the health facility. As a result, patients can develop positive attitudes towards the health facility and positive treatment intentions.

## 5. Conclusions

The reasons for treatment default in this sample are mainly socially derived, which confirms that social situations are determinants of health. These matters need to be considered by health care workers, and be integrated into communication packages with patients in similar settings. Failure to address these issues will continue to perpetuate treatment default, which frustrates treatment outcomes for both the patient and the health system. In particular, the cultural understanding of mental illness in traditional and mainly rural communities, and the practice of consulting traditional healers [37] for mental disorders, implies that health professionals cannot continue to ignore this aspect because it impacts treatment compliance among some people with mental disorders.

Although the use of technologies has been reported to increase treatment compliance among patients with mental disorders in low- and middle-income countries [39], the feasibility of this strategy cannot be assumed for rural communities in South Africa. This is because South Africa has its own extreme levels of poverty, especially in rural communities. Kwa-Zulu Natal, the setting of this study, has been identified as one of the provinces with high poverty levels in its rural areas [40], and mobile phones are not as easily available among the patients as in other communities.

## 6. Recommendations

The following recommendation emerge from the study:i.Both health literacy levels and cultural orientation should be specifically addressed as health professionals interact with mentally ill patients, especially in rural communities where general literacy, in addition health literacy, is often lower [33].ii.Although foreign to scientific medicine, health workers in such areas need to know about the understanding and explanation of certain mental illnesses. In a study conducted in the same province, the cultural conceptualization, understanding, and/or explanation of schizophrenia were found to be a supernatural phenomenon of *ukuthwasa* [41], which requires a cultural ritual or intervention by a traditional healer that, by default, is not recommended to take place at a health facility. The respect for this cultural view of mental illness is likely to contribute to health workers counseling patients to continue attending their treatment appointments at the health facility. Furthermore, a possible collaboration between traditional and medical practitioners may be forged, with the purpose of traditional healers utilizing their influence to encourage patients to attend health facility treatment.iii.Both health education and promotion for patients with mental disorders should be tailor-made according to the profile of the community from which the patients come.iv.It is recommended the findings of this study be used as a basis for a bigger quantitative study to explore specific demographic variables that are associated with treatment default.

## 7. Limitations of the Study

Because of its exploratory and qualitative nature, the sample size was too small for any meaningful quantitative analysis, even for demographic data. Because of its purposive nature, the sample was also recruited at a point in time, and the data may have been different if the study was conducted during a different period.

## Figures and Tables

**Table 1 healthcare-09-00461-t001:** Socio-demographic information.

Variables	Frequency	Percentages
**Gender**		
males	11	52.38%
females	10	47.62%
**Age**		
above 35	11	52.38%
below 35	10	47.62%
**Marital Status**		
Single	9	42.86%
Married	4	19.05%
Divorced	5	23.81%
Widowed	3	14.28%
**Level of Education completed**		
Primary education	12	57.14
Secondary education	7	33.33%
No formal schooling	2	9.53%
**Disability Grant**		
Receives a disability grant	8	38.10%
Does not receive a disability grant	13	61.90%
**Diagnosis**		
Schizophrenia	15	71.42%
Major Depressive Disorder	6	28.58%
**Use of psychoactive substances**		
Alcohol	10	47.62%
Dagga	4	19.04%
Whunga	1	4.76%
None	6	28.58%

## Data Availability

The authors are not able to share the data because of the promise they gave to the participants that they will report on the findings without sharing the raw data.

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
