# Peer review of "Why Do Patients with Mental Disorders Default Treatment? A Qualitative Enquiry in Rural Kwazulu-Natal, South Africa"

_healthcare, 2021, doi:10.3390/healthcare9040461_

Round 1

Reviewer 1 Report

Thank you for the opportunity to read this very interesting paper. The paper has many strengths that should be of interest to the journal audience. Thus, the following suggestions are around enhancing the presentation for publication and clarifying aspects of the data and reporting.

I will try to be as specific as possible in noting the area I am speaking about.

Abstract

The abstract should be a single paragraph and should follow the style of structured abstracts, but without headings.

Objective. Authors must specify the population.

  1. Materials and Methods

1st paragraph: When was the data collected? (Month/Year)

How was the sample chosen? Why was those 21 participants chosen and not others? Authors must specify it.

Which is the ID number of Ethics Committee? Interventionary studies involving animals or humans, and other studies require ethical approval must list the authority that provided approval and the corresponding ethical approval code. Please include the date and code register number of ethics committee.

Which are the principal themes in-depth interviews?

  1. Results

You must change Findings by Results

Sociodemographic characteristics. When you say… “Most participants were older than 35 years”. Authors must review this sentence because there is a little difference: 52.38% vs 47.62%. Mean (SD) could be better?

Are there only one “43 years old male” / “31 year old female” / …?

  1. Discussion

Limitations related with the type of methodology used. Authors must specify it.

References

In the text, reference numbers should be placed in square brackets [ ] and placed before the punctuation; for example [1], [1–3] or [1,3].

Some references are not correct. Author 1, A.B.; Author 2, C.D. Title of the article. Abbreviated Journal Name Year, Volume, page range.

I wish you all the best.

Author Response

Dear Editor

I am attaching responses to the comments of both reviewers.

Kind regards

Kebogile Mokwena

Reviewer 2 Report

Quite a few non-institutionalised patients suffering from mental disorders do not come to planned visits with their psychiatrists/therapists/mental health care centres and do not pick up/use recommended medication. Treatment default is worrying for several reasons. Mental conditions will go unattended, patients’ mental health problems which could be alleviated by treatment may worsen, health care providers/mental health institutions will lack feedback about the effects of their treatment and the conditions of their patients, efficiency in healthcare institutions deteriorates when scheduled appointments do not take place, etc.

This is the motivation for the present study which states that “there is a need to conduct studies in various settings to enable responsive interventions that can help mental health care service providers to understand the personal, social and health related issues encountered by mental health care users, that contribute to treatment defaulting” (p.2).

The aim of the present study is accordingly to investigate the reasons for treatment default and thereby to identify measures to reduce it. The setting of the study is a rural district in South Africa where treatment default appears to be a widespread phenomenon. The study used an “explorative qualitative design” with “in-depth interviews” of 21 patients, diagnosed with schizophrenia or major depressive order, who “had defaulted their monthly follow-up appointments” at least twice. These patients were directly approached in their homes and persuaded to come to the hospital for being interviewed. In the Results section, a short description of the composition of the sample (gender, education, etc.) is given, before the major reasons for treatment default which emerged in the interviews are outlined. These reasons were beliefs that one was cured/no longer sick; lack of support/help from family/neighbours etc. to attend the appointments; permanently/temporarily away from the district; denial of having serious mental disorder; inadequate communications/explanations from the therapists/mental health care workers; difficulties because of flaws in a “disintegrated hospital system”; and resort to traditional healers.  The Discussion section expands somewhat on these reasons, stating, for instance, that traditional healers may to some extent initiate effective psychosocial interventions.

The study addresses an important topic for mental health care, and the background discussion is adequate and sufficient. The data collection seems well done (interviews were voice recorded, transcribed and coded using the NVivo program) and ethical considerations (such as informed consent of participants) seem satisfactory. The study is generally clearly written. Results are not surprising, but interesting. A merit is that the study draws on data from a part of the world which is under-represented in scientific studies of mental health care services. The study sample is quite small, but large enough for an explorative study. The authors writes that saturation was attained, i.e., no further interesting material/other important reasons would probably emerge if even more interviews had been conducted.

Thus, this reviewer sees several positive attributes of this study: relevance, study design, and originality – not in theories discussed or in methods or results, but original in the sense that it contributes with an examination of a less studied setting. However, this reviewer suggests that the points below should be considered.

  1. The two final sections Conclusion and Recommendations are very short. This reviewer thinks that the authors should consider whether there are ways of improving these sections. A main conclusion seems to be that reasons for treatment default are “socially derived which confirms that social situations are determinants of health” (Conclusion p.7, similar references to social factors and social determinants are made in the Abstract). This may be generally true, but should be developed more and linked better to proposed recommendations. “Flaws in the hospital system” and lack of social support due to family dissolution are certainly “socially derived” reasons for treatment default, but what about denial? Proposed recommendations seem to be campaigns to raise “health literacy” and increase “cultural understanding of mental disorders”, but are these measures the only ones which emerge from the study? In general, this reviewer thinks the conclusion and recommendation sections are rather thin in content. They could be developed and refined, in more concrete ways. Perhaps also suggestions for needed future research could be added.

  1. There is a reference (on p.4) to a Table 2, but in the manuscript this reviewer received, no Table 2 could be found. It appears that Table 2 lists how many of the respondents who mentioned the different reasons for treatment default. The limited value of such a quantitative overview must be mentioned in the manuscript (the representativity of this small sample is unknown), but such an overview is interesting and should be included.

  1. Details/small comments: * The first references in the text is marked by [1,2], [3], etc, but after that, references are marked by superscript numbers? * MHCU (p.2) does probably mean “mental health care user” (?), but these four capital letters appears suddenly at p.2 without explanation. * P.3: Add information about the typical length (15 minutes? 30 minutes?) of the in-depth interviews?   *At the bottom of p.4 (subsection Lack of social …) – the disconnected words “may be other people in the family” should be removed?  * Table 1: Percentages reported with 2 digits are hardly necessary when the sample counts 21 individuals (nothing wrong in presenting these percentages, but they have little informational value in such a small sample with unclear representativity).  * Mostly, sentences are clear and easy to understand; still good idea to read critically through the final manuscript in order to avoid/remove diffuse sentences etc.

Author Response

Dear Editor

I am attaching our responses to the comments of reviewer 2

Regards

Kebogile Mokwena

Round 2

Reviewer 2 Report

In the opinion of this reviewer, the authors have made necessary and recommended changes in the revised manuscript. As regards the content, this reviewer finds the manuscript acceptable.

However, the manuscript should be read through in order to fix minor unclarities/spelling etc. Examples:

P.2, upper part of page:  “The special needs to care for such patients are often not met [10], which includes supervision for meeting hospital appointments, which is ongoing for maintaining optimum health.” Is the meaning of this sentence entirely clear?

P.2., middle part of page: “Mental health stigma discourages individuals from getting proper mental health treatment, and as the interface of mental illness, stigma, and mental health treatment has ethical and potentially moral implications, which cannot be ignored [16].” Something seems wrong in this sentence. Is this better: “Mental health stigma discourages individuals from getting proper mental health treatment, and the interface of mental illness, stigma, and mental health treatment has ethical and potentially moral implications which cannot be ignored [16].”

P.3, second line: Why  “mental health Unit”  - not “mental health unit”?

P.3, the word “fro” ?

P.3. Line 8 refers to “schizophrenia or major depressive disorder”, later on the same page capital letters are used: “Schizophrenia or Major Depressive Disorder”. Why the difference?

P.3, last line in Data collection section: “The interviews … was voice recorded..”  Should it be “interviews .. were voice recorded..”?

P.7, last line in Conclusion section: “Kwa-Zulu Natal, the setting of this study, has been identified as one of the provinces with high poverty levels in its rural areas [40], and mobile phones may not as easily avail-able among the patients as in other communities.” Should it be “..], and mobile phones are not as easily available among the patients…”

There is no need for further comments by this reviewer.
